# Polymorphism rs564398 of the ANRIL Gene as a Coronary-Artery-Disease-Associated SNP in Diabetic Patients of the Kazakh Population

**DOI:** 10.3390/diagnostics14212412

**Published:** 2024-10-29

**Authors:** Alisher Aitkaliyev, Nazira Bekenova, Tamara Vochshenkova, Balzhan Kassiyeva, Valeriy Benberin

**Affiliations:** 1Department of Science, Medical Centre Hospital of President’s Affairs Administration of the Republic of Kazakhstan, Mangilik El 80, Astana 010000, Kazakhstan; nazira.bekenova@mail.ru (N.B.); vochshenkova@gmail.com (T.V.); kassiyevabs@gmail.com (B.K.); valery-benberin@mail.ru (V.B.); 2Institute of Innovative and Preventive Medicine, Alikhan Bokeikhan Street, Building 1, Astana 010000, Kazakhstan

**Keywords:** ANRIL gene, coronary artery disease, polymorphism rs564398

## Abstract

Background/Objectives. A cardiovascular complication of type 2 diabetes mellitus like coronary artery disease is influenced by a complex interplay between environmental, phenotypic, and genetic factors. The genetic mechanisms in the development of this pathology are not established. This study aims to evaluate the association of polymorphisms rs1011970, rs62560775, and rs564398 from the 9p21.3 locus with coronary artery disease in diabetic patients of the Kazakh population. Methods. A total of 343 people participated in the case-control study: the control group consisted of 109 people with type 2 diabetes and coronary artery disease, while the control group included 234 people. Genotyping was performed using real-time PCR. Statistical analysis was carried out using Chi-square methods and calculating odds ratios (OR) with 95% confidence intervals (CI). Results. According to the results, only the rs564398 polymorphism of the ANRIL gene was associated with coronary artery disease (*p* = 0.04). The CC genotype increased the risk of developing coronary artery disease by more than 1.5 times (1.62 (1.02–2.56)), whereas the TT genotype reduced the risk of coronary artery disease (0.39 (0.17–0.91)). The remaining polymorphisms, rs1011970 and rs62560775, were not associated with coronary artery disease. Conclusions. Thus, this research further elicits the association of the ANRIL gene with cardiometabolic disease.

## 1. Introduction

Coronary artery disease (CAD) develops when the myocardium is insufficiently supplied with oxygen and nutrients due to damage to the coronary arteries. Type 2 diabetes mellitus (T2DM) is a major independent risk factor for CAD, associated at baseline with a 2–3-times increased incidence of various clinical forms, such as stable angina (chronic form) and myocardial infarction (acute form) [1]. Prospective studies have indicated that diabetic patients have a two- to four-fold propensity to develop CAD, which is caused by coronary stenosis in arteries that supply blood to the heart [2,3].

Diabetes and CAD arise from a complex interplay of environmental factors, genetics, and epigenetics. About 40% to 50% of CAD cases are attributed to genetic factors, which makes research into the genetic causes of CAD a focal point in understanding its pathogenesis [4]. Genome-wide association studies have identified genetic variants that contribute to the risk of CAD and T2DM at the 9p21 locus on chromosome 21, providing explanations for these findings [5].

The chromosomal locus 9p21.3, known as a genomic risk zone for cardiovascular diseases, includes two distinct haplotypes that confer risk for CAD and T2DM, which are widely distributed among different populations and account for 15% of cases of concurrent CAD and T2DM. These haplotypes consist of adjacent blocks of 50–100 single nucleotide polymorphisms (SNPs) separated by a recombination peak. They exhibit linkage disequilibrium, ensuring non-random co-inheritance for each disease [6]. As a basis for linking CAD and T2DM, researchers point to the potential overlap through long non-coding RNA ANRIL, a product of the cyclin-dependent kinase inhibitor (CDK2A/B) gene [7]. The identification of a potential transcriptional regulatory mechanism in this locus, induced by the long non-coding RNA ANRIL, suggests a common basis for CAD and T2DM [8].

Moreover, the direct vascular and immunomodulatory functions of the ANRIL gene activate systemic inflammation, accelerating signaling pathways (TNF-α-NF-kB-ANRIL and YY1-IL6/8), which exacerbates the development of cardiometabolic diseases [9].

The study includes three SNPs that are involved in the balance of cellular functions (cell cycle control, proliferation), linking diseases associated with the genetic locus 9p21.3: oncological, cardiovascular, and neurodegenerative diseases.

Thus, the potential common genetic signature for CAD and T2DM at the chromosomal locus 9p21.3 explains their development’s pleiotropic effect [10].

## 2. Materials and Methods

### 2.1. Study Design and Patient Selection

In total, 343 people participated in this case-control study. The control group with type 2 diabetes and CAD consisted of 109 people, while the control group included 234 people. All study participants were of Kazakh nationality. Patient recruitment took place in the therapeutic department of the Medical Centre Hospital of the President’s Affairs Administration of the Republic of Kazakhstan from September 2017 to August 2022. The control group was formed from individuals undergoing preventive check-ups at the same hospital. The data were collected retrospectively from clinical records and the hospital’s existing internal genotyping database.

The diagnosis of type 2 diabetes was established according to the American Diabetes Association (ADA, 2019) criteria [11]. The diagnosis of CAD was based on computerized tomography (CT) angiography.

Individuals in the control group were excluded from having diagnoses of diabetes and CAD based on historical, objective, and laboratory/instrumental data (glucose-level determination, electrocardiography).

The inclusion criteria for the case group were confirmed diagnosis, age 18 years and older, and Kazakh nationality. Exclusion criteria included genetic diseases in the person’s medical history; hypothyroidism or hyperthyroidism; arrhythmias; implantation of LVAD within the last 3 months; regular alcohol consumption (more than 80 mL/day); anemia (Hb < 110), cancer; kidney disease; severe cardiovascular disease; liver disease; terminal stage of hematopoiesis; autoimmune diseases affecting autonomic nerve fibers such as systemic lupus erythematosus; concomitant degenerative diseases (e.g., Parkinson’s disease or multiple system atrophy); medications affecting the heart rate such as beta-blockers, verapamil, diltiazem, amiodarone, or nitrates; and pregnant or lactating women.

The control group’s inclusion criteria were the exclusion of diabetes, CAD diagnoses, age 18 years and older, and Kazakh nationality. The exclusion criteria were analogous to those of the case group.

Demographic data including gender, age, height, weight, and ethnic background were obtained from the medical records of the study participants.

Fasting glucose levels were determined from venous blood samples. Blood samples were taken from the antecubital vein in the procedure room after a 12 h fast. Plasma was separated by centrifugation at 1000× *g* (4 °C) for 10 min. Plasma for further biochemical analysis was stored at −30 °C. The serum obtained after centrifugation was used for analysis on the same day as the blood draw. Levels of glucose, total cholesterol, triglycerides (TG), HDL cholesterol (HDL-C), and LDL cholesterol (LDL-C) were measured using an enzymatic method on an automated biochemical analyzer, the Architect s 8000, manufactured by Abbott Laboratories, Abbott Park, IL, USA.

### 2.2. Genotyping

The genotyping process utilized advanced OpenArray technology, which facilitates reactions in small volumes. Custom-designed OpenArray slides, each containing 3072 data points, were employed. Pre-extracted DNA samples were mixed with a reaction mixture in a 384-well sample plate for genotyping. Each sample required 3.0 μL of OpenArray Real-time master mix and 2.0 μL of DNA sample at a concentration of 50 ng/μL. The total volume per well was 5 μL, and each sample was duplicated. The plate underwent thorough mixing and centrifugation.

Probes were designed using the QuantStudio OpenArray AccuFill Plate Configurator, and dried assays were placed in specific through-holes on the genotyping plates. These plates were specially engineered to include two allele-specific probes, a minor groove binder, and two PCR primers, ensuring precise and accurate genotyping calls. OpenArray technology utilizes nanoliter fluidics and can accommodate up to 3072 through-holes in various configurations.

A plate setup file was generated to outline the protocol for the applied samples, including analysis details. This file was uploaded into the QuantStudio™ 12K Flex software (v1.5) to execute the experiment. The prepared chips were inserted into the QuantStudio 12K Flex instrument using disposable genotyping blocks. Real-time PCR microfluidic technology facilitated the amplification reaction. Data resulting from the amplification reaction were analyzed using online tools provided by the Thermo-Fisher Cloud service. Bioinformatic analysis results allowed for the categorization of the studied genes as homozygotes for the major allele, homozygotes for the minor allele, or heterozygotes.

### 2.3. Statistical Analysis

Quantitative data were presented as medians with upper and lower quartiles (Me (Q1, Q3)) and treated as continuous variables. Given that the distribution of quantitative variables deviated from normality (checked using the Shapiro–Wilk test), non-parametric tests were applied. Group differences in BMI, total cholesterol, LDL cholesterol (LDL-C), HDL cholesterol (HDL-C), triglycerides (TG), and glucose levels were assessed using the Mann–Whitney U test.

Qualitative data were assessed using the chi-squared test, with calculations for odds ratios and confidence intervals. Qualitative data were presented as frequencies and proportions. Gender (male/female) and the presence of outcomes or characteristics (yes/no) were dichotomized. A significance level of *p* < 0.05 was considered for determining statistically significant differences. Data analysis was performed using SPSS 26.0 statistical software.

Allele and genotype frequencies of gene polymorphisms between groups were compared using Pearson’s chi-squared test and odds ratios (ORs) with 95% confidence intervals (CIs). Comparisons of genotype and allele frequencies were checked against the Hardy–Weinberg equilibrium. Statistical calculations were conducted using the Genetic Expert calculator for genetic analysis (http://gen-exp.ru/calculator_or.php, accessed on 28 October 2024).

## 3. Results

### 3.1. Comparison of Clinical and Demographic Indicators of Patients with CAD and the Control Group

Based on the research, the average age of patients with diabetes complicated by CAD was significantly older than that of the control group (*p* < 0.001). In the CAD group, the number of men was higher, while women predominated in the control group. These gender differences were statistically significant (*p* < 0.001) (Table 1).

The BMI in the patient group was significantly higher than in the control group (*p* < 0.001). The glucose level in patients with CAD was almost twice as high as in the control group (*p* < 0.001). Regarding lipid metabolism indicators, statistically significant differences were found in triglyceride, HDL, and LDL levels. According to our data, the average triglyceride levels were higher in the CAD patient group (*p* < 0.001). HDL and LDL levels were significantly lower in patients with CAD than in the control group (*p* < 0.001 and *p* = 0.005, respectively). Although cholesterol levels were lower in patients compared to the control group, these differences were not statistically significant (Table 1).

### 3.2. The Prevalence of Alleles and Genotypes of Gene Polymorphisms Among Patients with CAD and Individuals in the Control Group

The study on the prevalence of rs1011970 alleles showed that the frequency of the G allele predominated in both the case group and the control group, with the T allele frequency being lower. Regarding genotype frequencies, they were also similar between both groups. The GG genotype was most observed (Table 2), while the heterozygous GT genotype was less frequent in both groups. The TT genotype was the least common (Table 2).

Among patients with CAD, the A allele of rs62560775 was more prevalent compared to the G allele (Table 2). Similarly, in the control group, the distribution of alleles was analogous, with the A allele predominating over the G allele. The frequency of genotype distributions was also comparable between the patient group and the control group. In both groups, the AA genotype was most frequently observed. The heterozygous AG genotype was less common. The GG genotype was rare, occurring very infrequently in both the CAD patients and the control group (Table 2).

Regarding the allele frequency of the rs564398 polymorphism, the C allele was more common than the T allele among both patients with CAD and in the control group. The CC genotype frequency was the highest among the case group and the control group (Figure 1). The CT genotype was less frequent, while the TT genotype was the least frequent among both groups (Table 2).

### 3.3. Association of 9p21.3 Locus Polymorphisms with CAD in Patients with T2DM

According to the results, only rs564398 was associated with CAD (*p* = 0.04). Specifically, the CC genotype increased the risk of developing the condition by more than 1.5 times, whereas the TT genotype reduced the risk of CAD. The remaining polymorphisms, rs1011970 and rs62560775, showed no association with CAD (Table 2).

## 4. Discussion

The study suggests the CC genotype of the rs564398 polymorphism predisposes Kazakh individuals with T2DM to the development of CAD. In contrast, the TT genotype reduced the risk of CAD development.

Though GWAS studies have consistently found connections between the genomic region containing ANRIL and the risk of developing T2DM and CAD, the genetic factors associated with these conditions at the ANRIL locus tend to be distinct, with atherosclerosis-related SNPs scattered throughout the ANRIL gene and T2DM SNPs located further away from the last exon of ANRIL [12]. However, Zeggini E. et al. found the SNP rs564398 to be an exception: this polymorphism is linked to both T2DM and CAD [13]. The results correlate with the recent study of rs564398 in association with CAD risk factors in Turkish patients (*n* = 1285): women with the rs564398 CC genotype showed increased susceptibility to CAD (*p* = 0.02) and severe CAD (*p* = 0.05) [14]. Additionally, that study reported that the T allele of rs564398 was more prevalent among hypertensive males, while the C allele for rs564398 was associated with a decreased risk of T2DM (*p* = 0.02) [14]. M.S. Cunnington previously reported the T allele of rs564398 was associated with the under-expression of ANRIL [15]. Interestingly, the downregulation of linear ANRIL has been suggested to correlate with reduced proatherogenic characteristics [14]. Therefore, individuals who do not carry the T allele of rs564398 may face an increased risk of CAD [14].

T2DM-associated CAD is a heterogeneous and polygenetic disorder with multifactorial pathogenesis, influenced by the interplay of different genes and the environment [16]. Recent GWAS have identified numerous genetic loci associated with T2DM-associated cardiovascular complications, with one of the most consistently significant loci across multiple populations being the ANRIL gene on chromosome 9p21.3 [17]. Genetic polymorphisms in the ANRIL gene have been implicated in developing and regulating lipid metabolism, nerve repair, and regeneration [18,19,20,21].

The SNP rs564398 is situated approximately 100 kb upstream of the CDKN2A/2B genes, which is comprised of two protein-coding genes and a long non-coding RNA known as CDKN2Β-AS (antisense to CDKN2B) or ANRIL: CDKN2B encodes p15^INK4B^ and tumor suppressor protein (INK)-4 protein p15^INK4B^, while CDKN2A encodes p16^INK4a^ [22]. These genes encode key tumor suppressor proteins regulating the cell cycle and TGF-β transformation, through which it may contribute to the pathogenesis of atherosclerosis [23].

The phenome-wide association (PheWAS) plot indicated the significant (*p* ≤ 0.05) associations of rs564398 for CAD, myocardial infarction, abnormal aortic aneurysm, T2DM, heart failure, leukemia, glaucoma, and esophageal squamous cell carcinoma [16,24,25].

The molecular manifestations of SNP rs564398 contribute to atherosclerosis development. Namely, two studies indicate that this SNP is strongly associated with ANRIL expression: it is predicted to interfere with the Ras-responsive element binding protein 1 (RREB1) binding site within the 9p21 locus [8,15]. While Ras oncogenes are widely recognized for their involvement in cancer development, they also play a role in atherosclerosis by promoting vascular aging and stimulating the expression of pro-inflammatory cytokines [8,15]. Moreover, S. Pechlivanis reported a significant correlation between rs564398 and coronary artery calcification (CAC), another indicator of CAD, noting that this link was predominantly driven by stronger effects in males [26].

Several limitations should be considered when interpreting the results of this study. The incompatibility of groups is the biggest shortcoming of our study. The sample size was relatively small, which could impact the generalizability of the findings. Secondly, the patients enrolled in this study were recruited under hospital-based circumstances, potentially limiting their representativeness of the broader diabetic population with CAD, in this case, the ethnically Kazakh population.

## 5. Conclusions

This study contributes valuable insights into linking the genetic underpinnings of the ANRIL gene and the risk of developing cardiometabolic disease, namely CAD within a Kazakh ethnic group. Our findings on the predisposition to CAD in individuals with the CC genotype of rs564398 complement and confirm the existing patterns in the disease’s pathogenesis, previously reported regarding the association of rs564398 with both T2DM and CAD in UK samples.

Therefore, SNPs of the genetic locus 9p21.3, particularly rs564398 of the ANRIL gene, involved in the balance of cellular functions (cell cycle control, proliferation), may contribute to the development of CAD in the context of T2DM and represent potential for further exploration of their application in early diagnostics and managing diabetes.

## Figures and Tables

**Figure 1 diagnostics-14-02412-f001:**
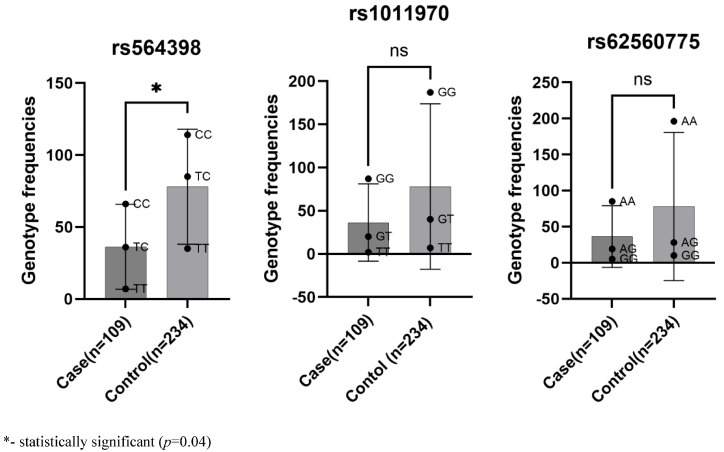
The frequency of genotypes of gene polymorphisms among patients with CAD and the control group.

**Table 1 diagnostics-14-02412-t001:** Anthropometric and clinical characteristics of 343 patients with type 2 diabetes.

	Case (*n* = 109)	Control (*n* = 234)	*p*
Age	58 (54–65)	53(49–57)	<0.001 ^b^
Male	76 (69.7%)	75 (32.1%)	<0.001 ^a^
Female	33 (30.3%)	159 (67.9%)	
BMI (kg/m^2^)	29.33 (26.9–32.1)	25.96 (23.3–30.1)	<0.001 ^b^
Glucose (mmol/L)	9.06 (7.41–11.7)	5.17 (4.87–5.44)	<0.001 ^b^
TG (mmol/L)	1.92 (1.48–2.83)	1.36 (0.92–1.57)	<0.001 ^b^
Total cholesterol	5.43 (4.45–6.29)	5.46 (4.85–6.10)	0.79 ^b^
Low-density lipoprotein (LDL)	3.10 (2.35–3.99)	3.48 (2.87–4.03)	0.005 ^b^
High-density lipoprotein (HDL)	1.06 (0.93–1.23)	1.39 (1.25–1.53)	<0.001 ^b^

^a^—comparisons were made using the chi-square test. ^b^—Mann–Whitney U test was used to compare median values.

**Table 2 diagnostics-14-02412-t002:** The prevalence of alleles and genotypes of gene polymorphisms among patients with CAD and the control group.

Polymorphisms	Alleles/ Geno-Types	Frequencies		*p*	OR (95%CI)	HWE	
		Case (*n* = 109)	Control (*n* = 234)			Case	Control
rs1011970	G	97 (88.9%)	207 (88.5%)	0.84	1.05 (0.63–1.76)	0.55	0.12
	T	12 (11.1%)	27 (11.5%)		0.95 (0.57–1.58)		
	GG	87 (79.9%)	187 (79.9%)	0.8	0.99 (0.56–1.75)		
	GT	20 (18.3%)	40 (17.1%)		1.09 (0.60–1.97)		
	GG	2 (1.8%)	7 (3.0%)		0.61 (0.12–2.97)		
rs62560775	A	95 (86.7%)	210 (89.7%)	0.24	0.74 (0.46–1.22)	0.003	0.14
	G	14 (13.3%)	24 (10.3%)		1.34 (0.82–2.20)		
	AA	85 (78.0%)	196 (83.8%)	0.38	0.69 (0.39–1.22)		
	AG	19 (17.4%)	28 (12.0%)		1.55 (0.82–2.93)		
	GG	5 (4.6%)	10 (4.2%)		1.08 (0.36–3.23)		
rs564398	T	25 (22.9%)	68 (29.1%)	0.007	0.60 (0.42–0.87)	0.65	0.06
	C	84 (77.1%)	166 (70.9%)		1.66 (1.15–2.41)		
	TT	7 (6.4%)	35 (15.0%)	0.04	0.39 (0.17–0.91)		
	TC	36 (33.0%)	85 (36.3%)		0.86 (0.53–1.40)		
	CC	66 (60.6%)	114 (47.8%)		1.62 (1.02–2.56)		

## Data Availability

The original contributions presented in the study are included in the article, further inquiries can be directed to the corresponding author.

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
