# Peer review of "Polymorphism rs564398 of the ANRIL Gene as a Coronary-Artery-Disease-Associated SNP in Diabetic Patients of the Kazakh Population"

_diagnostics, 2024, doi:10.3390/diagnostics14212412_

Round 1
Reviewer 1 Report
Comments and Suggestions for Authors
Alisher Aitkaliyev et al. reported polymorphism rs564398 of the ANRIL gene as a common genetic marker for the development of type 2 diabetes-associated coronary artery disease and cardiac autonomic neuropathy. However, I disagree with their conclusion that rs564398 is a common genetic marker. I concur with a previous review (PMID: 30087655), which classified this as a disease-associated SNP rather than a common genetic marker.
Additionally, could you clarify the meaning of the three dots in Figures 1 and 2? They seem quite confusing and need further explanation.
Since the results presented in the manuscript do not fully support the conclusions drawn by the authors, I recommend rejecting the paper.
Author Response
Dear Reviewer 1,
Thank you very much for your comments and recommendations! Thanks to your recommendations, our manuscript has drastically improved. Here are our point-by-point responses to your comments. Moreover, grammatical corrections and syntax adjustments were made to address the quality of the English Language.
|
Comment Number |
Reviewers Comment |
|
|
Alisher Aitkaliyev et al. reported polymorphism rs564398 of the ANRIL gene as a common genetic marker for the development of type 2 diabetes-associated coronary artery disease and cardiac autonomic neuropathy. However, I disagree with their conclusion that rs564398 is a common genetic marker. I concur with a previous review (PMID: 30087655), which classified this as a disease-associated SNP rather than a common genetic marker. |
|
Author’s Response |
|
|
We do agree that rs564398 is not a common genetic marker for type 2 diabetes-associated coronary artery disease and cardiac autonomic neuropathy. It is rather that the SNP rs564398 is a coronary artery disease-associated SNP. Therefore, we have made major manuscript revisions and corrected the study design and the conclusions. Cardiac autonomic neuropathy was removed altogether from the revised manuscript. The study design includes only patients with coronary artery disease (specifically in the context of diabetes) and a control group. |
|
|
|
Reviewers Comment |
|
Additionally, could you clarify the meaning of the three dots in Figures 1 and 2? They seem quite confusing and need further explanation. Since the results presented in the manuscript do not fully support the conclusions drawn by the authors, I recommend rejecting the paper. |
|
|
Author’s Response |
|
|
The figures 1 and 2 were removed from the revised manuscript as they do not bring the meaningful data that is not described in the main text or the tables. |
Reviewer 2 Report
Comments and Suggestions for Authors
Major points to note.
Study design. Why did the authors not compare the 2 groups, control and diabetes? The differences in the diabetes subgroups may be associated with the underlying disease rather than complications. Why did they not compare the CAN and CAD groups?
The authors should have matched donors of comparable age, not younger. Suddenly, they didn't ‘live to be diabetic.’ They should also have matched donors by gender. Otherwise, this is not consistent with a case-control type study.
What is MAF in the general population? Are there data, your own or from the literature? Are the results obtained consistent with them?
There are errors and typos, the English language needs revision.
Minor remarks.
Introduction. The authors write ‘The study includes three SNPs that did not show associations with cardiovascular 86 diseases (CVD) and TDM2...’ This is not entirely true. There is evidence in the literature of an association between polymorphism rs564398 and diabetes and CVD. The authors themselves also write about it in the Discussion section.
Methods. Lines 166 - 169 - the authors should indicate which method they used to assess differences in qualitative variables.
The authors should check the p-value, especially in clinical data, it is very doubtful that for not very large differences (in age, for example) p<0.001.
Results. The authors write ‘Based on the research, the average age of patients with diabetes complicated by CAD was significantly lower than that of the control group (58(54-65) and 53(49-57), p<0.001).’ The figures suggest that they were older, not younger
Data in tables and text should not be repeated.
Table 1 caption. Medians rather than mean values were compared.
Table 2. For what purpose is the χ2 value given?
Where is Table 3 with the CAN group data? Subsection numbering is broken, section 3.3 comes after 3.5. Section 3.4. occurs in two places. There is no coherent logic in the presentation of the results + there is repetition of data.
In the text, there are variants of abbreviations T2DM/T2D and CAN /DCAN, the article needs uniformity.
Line 116 ‘...regular alcohol consumption (more than 80 mg/day)...’ Is this a significant amount of alcohol? Maybe ml?
Line 135-136. The authors write ‘Body Mass Index (BMI) was calculated by dividing weight in kilograms by the square of height in metres.’ It doesn't need to be explained if it's an accepted method.
If there is no significant difference, it is better to write ‘ns’ rather than giving a p-value.
The list of abbreviations, in my opinion, is unnecessary. Some of the abbreviations in the list do not occur in the text.
Author Response
Dear Reviewer 2,
We are very grateful for your review of our work! We believe that your comments and recommendations will greatly enhance the quality of our article. Your suggestions have helped us gain significant clarity on the mechanisms and pathogenesis of the disease. Thanks to your clear review, we have re-evaluated many aspects of our work and made substantial revisions. This has been a valuable experience for us in acquiring new knowledge and skills!
Grammatical corrections and syntax adjustments were made to address the quality of the English Language.
The given lines in the revised manuscript are indicated in a “simple mark-up” form of tracked changes.
|
Comment Number |
Reviewers Comment |
|
|
Study design. Why did the authors not compare the 2 groups, control and diabetes? The differences in the diabetes subgroups may be associated with the underlying disease rather than complications. Why did they not compare the CAN and CAD groups? |
|
Author’s Response |
|
|
Following the recommendation of the first reviewer, we revised the study design to include only patients with coronary artery disease (specifically in the context of diabetes) and a control group. |
|
|
|
Reviewers Comment |
|
The authors should have matched donors of comparable age, not younger. Suddenly, they didn't ‘live to be diabetic.’ They should also have matched donors by gender. Otherwise, this is not consistent with a case-control type study. |
|
|
Author’s Response |
|
|
Unfortunately, the incompatibility of groups is one of the major shortcomings of our study, which we noted in the limitations of the study in the Discussions section – lines 219-220. However, considering that the data were collected retrospectively from clinical records and the Hospitals existing internal genotyping database, we can reformulate the sample to ensure comparability and recalculate the data if necessary. The data collection has also been included in the methods section – lines 70-72. |
|
|
|
Reviewers Comment |
|
What is MAF in the general population? Are there data, your own or from the literature? Are the results obtained consistent with them? |
|
|
Author’s Response |
|
|
According to PubMed SNP data, the MAF of the rs564398 polymorphism for the C allele is 21% in the general population. However, our data on the prevalence of the C allele in the healthy population (control group) contradicts the SNP database, showing a rate of 70.9%. |
|
|
|
Reviewers Comment |
|
There are errors and typos, the English language needs revision. |
|
|
Author’s Response |
|
|
The manuscript has undergone major English language revisions, and the sentences are clearer, and errors were corrected. |
|
|
|
Reviewers Comment |
|
Introduction. The authors write ‘The study includes three SNPs that did not show associations with cardiovascular 86 diseases (CVD) and TDM2...’ This is not entirely true. There is evidence in the literature of an association between polymorphism rs564398 and diabetes and CVD. The authors themselves also write about it in the Discussion section. |
|
|
Author’s Response |
|
|
The sentence has been amended. Now it reads “The study includes three SNPs that are involved in the balance of cellular functions (cell cycle control, proliferation)...” – line 58-59. |
|
|
|
Reviewers Comment |
|
Methods. Lines 166 - 169 - the authors should indicate which method they used to assess differences in qualitative variables. |
|
|
Author’s Response |
|
|
Qualitative data were assessed using the Chi-square test, with calculations for odds ratios and confidence intervals, - line 126. |
|
|
|
Reviewers Comment |
|
The authors should check the p-value, especially in clinical data, it is very doubtful that for not very large differences (in age, for example) p<0.001. |
|
|
Author’s Response |
|
|
We double-checked the p-values. Unfortunately, significant differences were characteristic in the comparison of the data. |
|
|
|
Reviewers Comment |
|
Results. The authors write ‘Based on the research, the average age of patients with diabetes complicated by CAD was significantly lower than that of the control group (58(54-65) and 53(49-57), p<0.001).’ The figures suggest that they were older, not younger. |
|
|
Author’s Response |
|
|
Thank you very much for the correction. We indeed made an error and wrote "younger". We have corrected this in the text, - 138-139. |
|
|
|
Reviewers Comment |
|
Data in tables and text should not be repeated. |
|
|
Author’s Response |
|
|
Thank you for your recommendations! We have removed the data from the text that are indicated in the table. |
|
|
|
Reviewers Comment |
|
a) Table 1 caption. Medians rather than mean values were compared. b) Table 2. For what purpose is the χ2 value given? c) Where is Table 3 with the CAN group data? Subsection numbering is broken, section 3.3 comes after 3.5. Section 3.4. occurs in two places. There is no coherent logic in the presentation of the results + there is repetition of data. |
|
|
Author’s Response |
|
|
a) Thank you very much for your comment! We have corrected "mean" to "median" in the table notes. b) When presenting results in a table, it is sometimes recommended to include the Chi-square value. However, we completely agree with you and have decided that it is unnecessary to show this value. Thank you for your comment! c) The Table 3 was deleted from the manuscript as CAN values were removed from the manuscript altogether. |
|
|
|
Reviewers Comment |
|
In the text, there are variants of abbreviations T2DM/T2D and CAN /DCAN, the article needs uniformity. |
|
|
Author’s Response |
|
|
The uniformity of abbreviations was obtained throughout the amended manuscript. |
|
|
|
Reviewers Comment |
|
Line 116 ‘...regular alcohol consumption (more than 80 mg/day)...’ Is this a significant amount of alcohol? Maybe ml? |
|
|
Author’s Response |
|
|
Thank you for your comment! Indeed, it should be in "ml" – now the text reads “…regular alcohol consumption (more than 80 ml/day).” |
|
|
|
Reviewers Comment |
|
Line 135-136. The authors write ‘Body Mass Index (BMI) was calculated by dividing weight in kilograms by the square of height in metres.’ It doesn't need to be explained if it's an accepted method. |
|
|
Author’s Response |
|
|
Thank you for your comment! This sentence has been removed from the amended manuscript. |
|
|
|
Reviewers Comment |
|
The list of abbreviations, in my opinion, is unnecessary. Some of the abbreviations in the list do not occur in the text. |
|
|
Author’s Response |
|
|
The list of abbreviations has been removed from the text. |
Round 2
Reviewer 1 Report
Comments and Suggestions for Authors
Thank you for agreeing to make the changes and considering my comments.
However, I believe the original Figures 1 and 2 are meaningful, even though they don't show three dots. You displayed many individuals in each group without explaining the meaning of the dots—whether they represent the highest, lowest, or median values. Including this explanation would help support your conclusion, so you can't just remove the data.
Comments on the Quality of English Languageclear.
Author Response
Dear Reviewer 1,
Thank you very much for your comments and recommendations!
|
Comment Number |
Reviewers Comment |
|
|
Thank you for agreeing to make the changes and considering my comments. However, I believe the original Figures 1 and 2 are meaningful, even though they don't show three dots. You displayed many individuals in each group without explaining the meaning of the dots—whether they represent the highest, lowest, or median values. Including this explanation would help support your conclusion, so you can't just remove the data. |
|
Author’s Response |
|
|
Thank you for your response. The Figure 1 has been put back to the manuscript with the indication of what the dots mean. Figure 1 is the visual representation of the frequency of genotypes of gene polymorphisms among patients with CAD and the control group, and each dot denotes a frequency of a particular genotype. For example, the frequency of genotype CC of rs564398 is 66 in case group, which is indicated on the top of the Y-axis. The figure is a visual illustration of the data that is provided in table 2.
However, we do agree that further explanation of our results is needed for our research, thus we have revised our discussion section – the main findings are described in the first paragraph lines 177-194. Moreover, we have revised the entire conclusion section to align with the key findings of our research – lines 227-236. |
Reviewer 2 Report
Comments and Suggestions for Authors
The authors have substantially revised the article—in fact, they have written a new article. They considered my comments and, apparently, those of other reviewers.
Author Response
Dear Reviewer 2,
thank you for your review and valuable insights. We will follow your comments and recommendations about the study design in our future research. We deeply appreciate your feetback!
Kind regards,
Alisher Aitkaliyev